# ICNN: INPUT-CONDITIONED FEATURE REPRESENTATION LEARNING FOR TRANSFORMATION-INVARIANT NEURAL NETWORK

## ABSTRACT

We propose a novel framework, ICNN, which combines the input-conditioned filter generation module and a decoder based network to incorporate contextual information present in images into Convolutional Neural Networks (CNNs). In contrast to traditional CNNs, we do not employ the same set of learned convolution filters for all input image instances. And our proposed decoder network serves the purpose of reducing the transformation present in the input image by learning to construct a representative image of the input image class. Our proposed joint supervision of input-aware framework when combined with techniques inspired by Multi-instance learning and max-pooling, results in a transformation-invariant neural network. We investigated the performance of our proposed framework on three MNIST variations, which covers both rotation and scaling variance, and achieved 0.98% error on MNIST-rot-12k, 1.12% error on Half-rotated MNIST and 0.68% error on Scaling MNIST, which is significantly better than the state-of-the-art results. Our proposed model also showcased consistent improvement on the CIFAR dataset. We make use of visualization to further prove the effectiveness of our input-aware convolution filters. Our proposed convolution filter generation framework can also serve as a plugin for any CNN based architecture and enhance its modeling capacity.

## 1 INTRODUCTION

Deep learning has been used to obtain promising results for various machine learning applications in recent times. Computer Vision has been at the frontier, driving its success with the introduction of Convolutional Neural Networks (CNNs). Face Recognition Li et al. (2015), Image Classification Deng et al. (2009), Action Recognition Chéron et al. (2015), and Speech AnalysisAbdel-Hamid et al. (2014); Yenigalla et al. (2018) are some of the areas which use CNNs extensively.

LeNet, first introduced by Lecun et al. (1998) served as the prototype for the modern-day Convolution Neural Network such as VGGNet Simonyan & Zisserman (2014) and ResNet He et al. (2015). These networks first consist of several layers of convolution, activation, and subsampling layers. The convolution filters are learned to extract local features from the image patches, which are subsampled using either max-pooling or average-pooling. Pooling reduces feature map resolution and sensitivity of the output to shifts and distortion. These local features are further combined hierarchically to form high-level features. The high-level features are then fed to the fully connected layers to form a compressed 1-D feature vector, which is then fed to a soft-max layer for Image Classification. These network uses techniques like data augmentation Fawzi et al. (2016) and auxiliary classification Yin & Liu (2018) to surpass many pre-established boundaries in Image Classification.

Although there has been a significant improvement in image classification, local features extracted by learned filters keep information regarding neighboring pixels of the image patch, irrespective of its location in the image. This makes networks prone to errors when they encounter images having variations like rotation, scaling, etc., as this requires networks to withhold complete structural information of the image up until the soft-max layer. Designing neural networks robust to such variations have been under research recently, resulting in many techniques which help propagate structural information of the given image till classification.

Data augmentation Fawzi et al. (2016) is a widely used approach which artificially increases the training data by incorporating different affine transformations, adding noise, scale changes to the images present in the original dataset based on prior knowledge. TI-POOLING Laptev et al. (2016) utilizes parallel Siamese architectures along with different transformation sets, inspired by multi-instance learning, and pooling operator, whereas MINTIN Zhang et al. (2018) uses Input Normalization and combines it with Maxout operator to tackle these variations. Image features like, SIFT Lowe (2004) has been used to extract features from key points, which are invariant to scale and rotation of the image.

A generic framework has been proposed in this work for generating input-conditioned convolutional filters along with a decoder network which serves as a generator module for learning representative images, which incorporates contextual information present in images into CNN models. In distinction with traditional CNNs, our framework generates a unique set of input-aware filters based on the given image, and thus enables the network to improve its modeling capacity. We also employ a decoder network which given the abstract features of an input image tries to construct a transformation free representative image of the input class.

We combined our proposed framework with techniques inspired by multi-instance learning Wu et al. (2015) and max-pooling operator Boureau et al. (2010) to generate transformation-invariant convolutional neural network feature representations. Unlike data augmentation, our model does not treat the original and its transformed instances as independent samples; we generate high-level representation from all the instances followed by a max-pooling layer (element-wise maximum of all representations) which are then passed as an input to the further layers. Combination of our input-aware filters, Convolution layers, element-wise maximum over feature representations, and Decoder network makes the final representation independent from the variations.

Implementation of this network topology makes use of parallel Siamese networks which is a combination of our proposed filter generator and decoder framework, convolution, pooling, and dense layers. We kept the convolution, pooling and dense layers consistent with those in Laptev et al. (2016); Zhang et al. (2018) to prove the effectiveness of our proposed input-aware framework. This topology supports weight sharing and takes a set of transformed image instances as input. We analyzed the performance of our proposed transformation-invariant network on variations of 2 datasets and achieved the error rate of 0.98% on MNIST-rot-12k Larochelle et al. (2007), 1.12% on half-rotated MNIST Jaderberg et al. (2015) and 0.68% on scaling MNIST which beats the state of the art accuracies as mentioned in Zhang et al. (2018). We also achieved consistent improvement on CIFAR dataset when compared with the baseline methods. Thus, validating the effectiveness of our proposed approach to handle transformations present in the images.

The following are the main contributions of this work:

- The first attempt, to the best of our knowledge, to make use of a decoder network to construct a representative image of the input class in order to mitigate transformations present in the input image.

- We also proposed run-time generation of input-conditioned convolution filters in our model architecture.

- The framework proposed in this paper can easily be integrated with any CNN based network to enhance its ability to incorporate contextual information present in images.

## 2   RELATED WORK

Recent research to acquire transformation invariance in convolutional neural networks have led to four main techniques. This includes Data Augmentation, TIPOOLING, MINTIN, and usage of transformation-invariant features like SIFT. David G. Lowe proposed SIFT Lowe (2004) to extract distinctive invariant features present in images, which can be used for matching different object or scene views reliably. The method required selection of key points in the image and generation of features robust to local affine transformation by SIFT descriptors. However, designing these general-purpose transformation-invariant features is a tedious and expensive process, and they can only handle very specific variations in the original data.

Data Augmentation increases the number of training samples in the original dataset by artificially introducing variations like rotations, scale changes, random crop, etc., based on prior knowledge. Although CNNs modeling capacity is flexible enough to support an increase in data samples, it results in increased training time and heavy usage of computational resources. The substantial increase in the number of parameters, necessary to tackle variations, can make a network prone to over or under-fitting.

TI-POOLING Laptev et al. (2016) introduced transformation invariance in a neural network using multiple instance learning (MIL) and pooling operator together. The proposed network consists of parallel Siamese network with transformed images as input. TI-POOLING operator is then applied to do max-pooling over the transformations from the parallel network. The TI-POOLED transformation invariant features are then fed to soft-max layer for image classification.

MINTIN Zhang et al. (2018) uses an input normalization module, to precalibrate the input image using prior knowledge, in combination with Maxout operator which can be employed into any CNN based architecture to address variations like rotation and scaling.

## 3 Approach

The proposed architecture employs input-conditioned convolutional filters to extract features from an input image. Framework for feature extraction from input images consists of two modules: Input-conditioned Filter Generation, which generates a unique set of filters given an input image, and a Convolution Module, which applies the generated set of filters on the input to extract context-dependent feature maps. This framework, Figure 2, is briefly explained below.

Before passing input images to the proposed feature extraction framework, we make use of transformations as described in Laptev et al. (2016). A predefined transformation set $\Phi$ is used to generate a collection of transformed instances of the given input image $\phi(x), \phi \in \Phi$. Figure 1 shows parallel instances of partial Siamese Networks consisting of our proposed feature extraction framework, two traditional convolutional (3x3 kernel size, kernels = 80, 160 respectively) and pooling (2x2 kernel size, stride = 2) layers, with ReLU as the activation function, which take the generated set of transformed image instances as input to produce high-level feature representations. We kept the same hyperparameter settings as in Laptev et al. (2016); Zhang et al. (2018) to show the improvement in adding our proposed framework. Following the notation of Zhang et al. (2018), we also denote high-level feature vectors by $f_\theta(x)$, where $x$ and $\theta$ denote the given input and all the parameters of the proposed modules, kernels and weights respectively. Thus for $n$ transformed input instances, we will obtain $n$ feature vectors denoted by $f_\theta(\phi_i(x)), i = 1, 2, ..., n$. After extracting high-level feature vectors, we apply element-wise maximum on all $f_\theta(\phi_i(x))$, which makes the output $g_\theta(x)$ either transformation invariant, in many cases, or at least makes it less dependent on the variations according to the Lemma 1 defined in Wu et al. (2015) and Laptev & Buhmann (2015).

$$g_\theta(x) = \max_{\phi_i \in \Phi} f_\theta(\phi_i(x))$$

We hypothesized that the generated output representation $g_\theta(x)$ is an abstract depiction of the input image as it takes the max of all the feature vectors generated from different variations of the image. Therefore, these features should have the sufficient information required to construct the image without any unnecessary transformation. Following this intuition, we introduced a convolution-based decoder network which takes this abstract representation as an input and tries to construct an image without any transformation.

Output representation $g_\theta(x)$ is also passed to dense layer, which makes use of ReLU activation function and Dropout technique, followed by a final output layer with softmax activation function. This deep learning architecture employs the joint supervision, $L_{total}$ of the Cross-Entropy, $L_{CE}$, and reconstruction loss, $L2 - loss$, and uses Adam optimizer. The $\lambda$ parameter is tuned as part of the training process.

$$L_{total} = L_{CE} + \lambda * L_{reconstruction-error}$$

Siamese networks make use of weight sharing among parallel instances, which leads to a drastic reduction in memory usage. In other words, memory footprint of the network is not directly related to the number of parallel instances in the Siamese network.

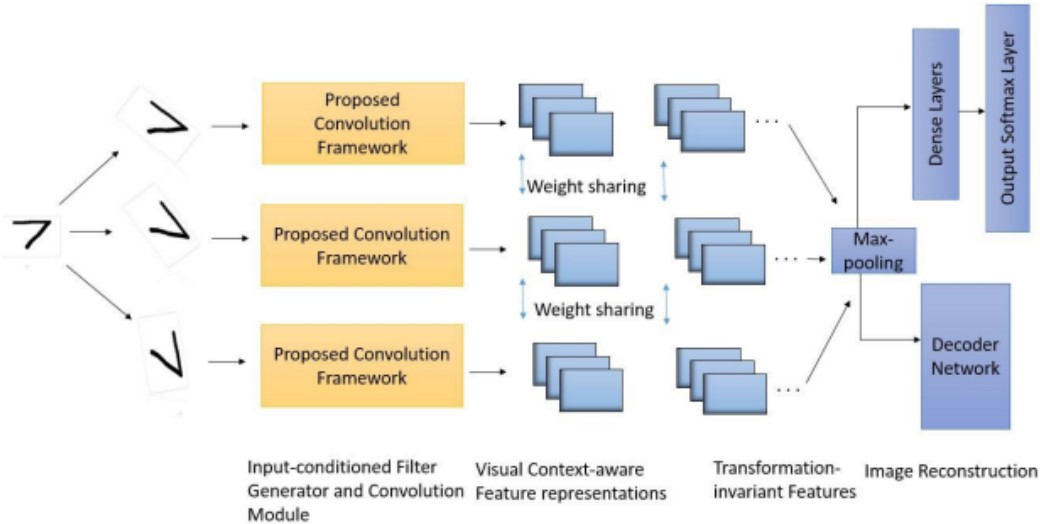

Figure 1: ICNN: Model Architecture

**Input-conditioned filter generation**: In theory, this generator module can be implemented as one of any deep differentiable functions. However, due to the remarkable success of convolutional neural networks in computer vision related tasks Deng et al. (2009); Simonyan & Zisserman (2014), we decided to make use of convolution based operations to implement this module. Given an input image X, it is first passed to a Convolution layer followed by a Max-pool layer to decrease the spatial size of the generated feature map. This feature map is then passed to a Deconvolutional layer, which performs fractionally-strided convolutions [17], to generate a distinctive set of filters. To remain consistent with the architecture in Laptev et al. (2016); Zhang et al. (2018), we fix the kernel size and number of kernels of the input-aware filters to be 3x3 and 40 respectively. We experimented with kernels of different sizes for the convolution and deconvolution layers to work with the above described input-aware filters hyperparameters and selected 3x3 and 2x2 respectively. We hypothesized that the use of different kernel sizes in deconvolutional layers will result in the capture of different level features.

**Convolution module**: A set of generated filters and an image sample is passed as an input to the convolution module. This module replicates the work of traditional convolutional layer and generates an encoded representation of the input image by convolving it with the given set of filters, followed by a pooling layer. No bias term is used in this module. The proposed filter generator and convolution module, in basic terms, is a generalized version of a standard convolution layer that can be represented as our proposed framework by generating the same set of filters independent of the visual context present in the input image. Our input-aware filter generation framework focuses on encoding visual context information into the convolutional filters that in turn generates context-dependent feature maps when convolved with an input image. In principle, the proposed filter generation framework can be easily combined with any CNN based architecture to incorporate visual context information.

**Decoder network**: The generated output representation $g_\theta(x)$ is fed in to the convolution based decoder network. Specifically, the decoder network serves the purpose of a generator and utilizes a combination of deconvolution, pooling, and convolution layers to construct the representative image of the input image. The use of a decoder network followed the intuition that by learning to construct transformation free image in parallel to classification, will help in producing a more generalized set of features and thus reducing the chances of overfitting. Further, due to the usage of the decoder model, the output representation $g_\theta(x)$ will be forced to learn those features that can construct the representative image with no transformations. Representative image of every class has been fixed before the training by analyzing the data corpus. For instance, in the case of MNIST dataset, we

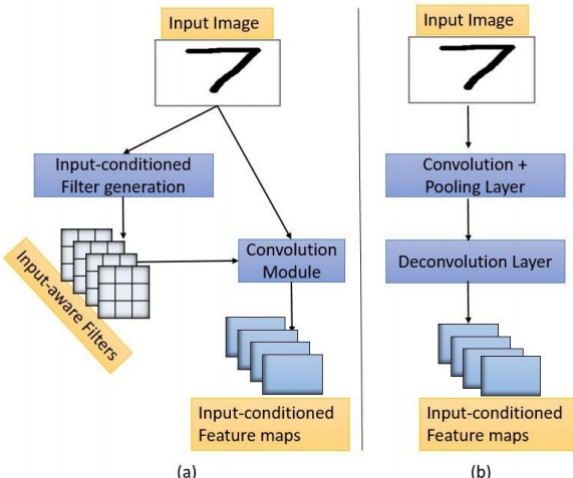

Figure 2: (a) Proposed convolution module (b) Detailed architecture of input-conditioned filter generation

selected a single image corresponding to every digit. This method can be easily extended to include multiple representative images of a single class in the training process. Further, to validate the effectiveness of the decoder network, we selected a random set of images from the test set and observed that the constructed images have less transformation when compared with the input image. Figure 3 presents a sample input and its corresponding constructed image to clearly depict the reduction in the image with the use of the decoder network. We removed blurriness using some artificial tools to make images more clear.

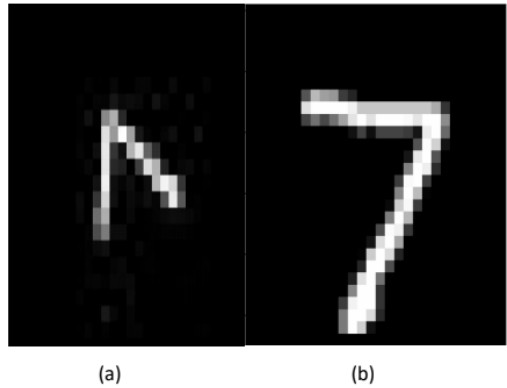

Figure 3: (a) Input test image (b) Constructed image using the Decoder network

## 4 EXPERIMENTS AND DISCUSSION

We analyzed the performance of our proposed network and presented the results of three computer vision datasets. MNIST dataset LeCun & Cortes (2010) variations designed to test rotation and scaling variations have been used for generating experimental results and are compared with state-of-the-art results presented by Zhang et al. (2018). We also make use of artificially rotated and scaled CIFAR-10 dataset for generating experimental results. To validate the ability to handle different variations, we kept our network topology same for different datasets. We have the same number of convolutions and dense layers as compared to Laptev et al. (2016); Zhang et al. (2018) in order to validate the performance of our proposed convolution framework. For all the experiments, 10% of

the training data is separated and used for validation. Also, all the reported results were aggregated after multiple runs.

**MNIST-Rotation-Variants**: MNIST LeCun & Cortes (2010) is a widely used toy dataset to study modifications proposed in computer vision algorithms. Two MNIST variations: MNIST-rot12k Larochelle et al. (2007) and Half-rotated MNIST Jaderberg et al. (2015) exist to analyze the performance of networks designed to be invariant to some particular transformations like rotations. As mentioned earlier, we used the same topology to check the robustness of our network but with a slight change in transformation set so as to remain consistent with prior research.

**MNIST-rot-12k** is the most commonly used dataset to validate proposed networks with rotation-invariance. It is composed by rotating original MNIST images by a random angle from $0^0$ to $360^0$. It contains 12000 instances for training and 50000 for testing. We analyzed the performance of our proposed model on three different transformations set $\Phi$ to compare results with Zhang et al. (2018). We make use of Dropout layer Srivastava et al. (2014) to reduce overfitting and train our network for around 50 epochs, being consistent with prior research.

Table 1 presents our experimental results on this dataset where channels denote the size of the transformation set used for a single input instance. We achieved consistent improvement when compared with other state-of-the-art approaches. Our proposed model achieves 0.98% error, while best approach in Zhang et al. (2018) achieve 1.57% error. It shows that our proposed context-aware convolution filters lead to significant improvement when handling rotation variance present in data.

**Half-rotated MNIST** is proposed by Jaderberg et al. (2015), mainly due to the small size of MNIST-rot-12k and to address the limitation of images being rotated by more than $90^0$. This dataset contains 60000 training samples and 10000 testing samples which are generated by taking original MNIST dataset and rotating them by a random angle from $-90^0$ to $+90^0$. This makes dataset more close to practical rotation variation scenarios. We achieved 1.12% error with our proposed network, where the model is trained for approximately 50 epochs, while best network in Zhang et al. (2018) achieves 1.23% error. The final experimental results and comparison on this dataset are given in Table 1.

**Scaling MNIST**: MNIST variations discussed above mostly validates rotation invariance. Therefore, to investigate the performance of our proposed approach on other modifications, such as scaling, we decided to introduce scaling variation as described in Zhang et al. (2018) artificially. Experimental results presented in Table 2 shows a significant improvement in error rate when compared with other state-of-the-art results.

**CIFAR-10** The publicly available CIFAR-10 dataset consists of 60,000 32x32 color images used for object recognition. There are 50,000 training images and 10,000 test images. Each image is assigned to one among ten classes. We artificially introduce rotation following both the strategies employed in generating MNIST-rot-12k and Half-rotated MNIST. This results in two rotated versions of CIFAR-10. We make use of TI-POOLING and MINTIN network as our baseline for these experiments. We observe that introduction of proposed convolution framework improves the error rate to 4.3% from 6.4% for rot-12k version, and to 5.3% from 7.1% for Half-rotated version.

We also experimented with network topologies consisting only multiple layers of our proposed convolution module in combination with pooling and dense layers, without any standard convolution layer, and achieved similar performance. This indicates that a single layer of our proposed convolution module can generate adequate transformation-invariant feature representations and incorporate contextual information present in images, making this module ideal to be used as a first-layer plugin in any existing CNN network. Similar improvements were observed in the case of scaling CIFAR dataset.

**Filter visualization**: To show the effectiveness of our input-conditioned convolution filters, refer to Figure 4, we use t-SNE visualization of the input-conditioned filters computed by the filter generation module for images within the test dataset. We randomly selected 200 samples from each class of MNIST-rot-12k test dataset for visualization. It can be easily noticed that filters with the same label make well defined and compact clusters proving our point that for the different input image, distinct filters are generated for better high-level feature extraction.

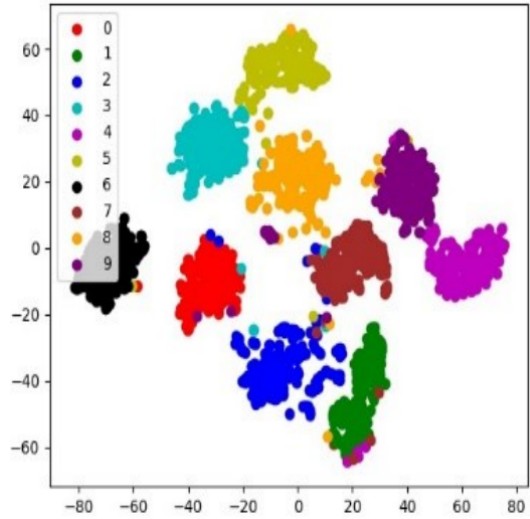

Figure 4: t-SNE visualization of filter weights

Table 1: Error on various rotated data corpus (%)

| Corpus | Approach | Channel=4 | Channel=7 | Channel=8 | Channel=13 | Channel=24 |
|--------|----------|-----------|-----------|-----------|------------|------------|
| MNIST-rot-12k | TI-POOLING | 2.47 | - | 1.88 | - | 1.61 |
| | MINTIN | 1.76 | - | 1.59 | - | 1.57 |
| | ICNN | **1.35** | - | **1.12** | - | **0.98** |
| CIFAR-10-rot-12k | TI-POOLING | 8.7 | - | 8.2 | - | 7.3 |
| | MINTIN | 7.4 | - | 6.8 | - | 6.4 |
| | ICNN | **5.3** | - | **4.5** | - | **4.3** |
| Half-rotated MNIST | TI-POOLING | - | 1.44 | - | 1.46 | - |
| | MINTIN | - | 1.32 | - | 1.23 | - |
| | ICNN | - | **1.18** | - | **1.12** | - |
| Half-rotated CIFAR-10 | TI-POOLING | - | 8.5 | - | 7.9 | - |
| | MINTIN | - | 7.2 | - | 7.1 | - |
| | ICNN | - | **5.9** | - | **5.3** | - |

## 5 CONCLUSION

We introduced a novel input-conditioned convolution filter framework, which unlike traditional CNNs, generates a different set of input-aware convolution filters conditioned on an input image instance. We also proposed a decoder network to mitigate the transformation present in the input images with the help of a set of pre-defined representative images of the input classes. We employed

Table 2: Error on various scaled data corpus (%)

| Corpus | Approach | Channel=5 | Channel=9 |
|---|---|---|---|
| Scaling MNIST | TI-POOLING | 1.52 | 1.32 |
| | MINTIN | 1.01 | 0.96 |
| | ICNN | **0.69** | **0.68** |
| Scaling CIFAR-10 | TI-POOLING | 7.1 | 6.8 |
| | MINTIN | 6.7 | 6.3 |
| | ICNN | **5.6** | **4.8** |

our proposed framework in combination with parallel Siamese networks and max-pooling technique to present transformation-invariant neural network. Our proposed filter generator is, in principle, a generic representation of a standard Convolution layer, which gives our proposed transformation-invariant network greater modeling flexibility. We investigated our proposed approach on the variations of 2 computer vision datasets, which covers both rotation and scaling variations, and achieved state-of-the-art results. Our proposed framework can also be easily integrated with any CNN model to enhance its ability to incorporate contextual information present in images.

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
