# OpenReview forum: "ICNN: INPUT-CONDITIONED FEATURE REPRESENTATION LEARNING FOR TRANSFORMATION-INVARIANT NEURAL NETWORK"
_ICLR.cc/2020/Conference — Reject_

### Official Review · AnonReviewer3 · 2019-10-19
**Official Blind Review #3**

**Rating:** 1

**Review:**

*Paper summary*

The authors build an input transformation invariant CNN using the TI-Pooling architecture of (Laptev et al., 2016). They make some modifications to this, namely 1) they replace the convolutional filters, with input-dependent convolutional filters, and 2) they add a decoder network from the final representation, which reconstructs an input transformation rectified image, to encourage the final representation to be fully transformation invariant.

*Paper decision*

I have decided to assign this paper a reject because of two main reasons.

*Supporting arguments*

One reason is that the base architecture is not novel. This in itself is not a key issue, but I would expect the authors to have done some in depth analysis or experimentation otherwise to compensate for this. I regret, the authors may just not have known that the ideas were already explored in the literature. The second reason is that the work is not well placed in context with prior works. This is both evident in the lack of referenced works (see below for a list) and the lack of sufficient baselines, against which they compare. For instance, if the authors had considered “Learning Steerable Filters for Rotation Equivariant CNNs” by Weiler et al. (2018) they would have known that their MNIST-rot-12k results are not state of the art as they state. In Weiler et al., the authors report 0.714 test set error on MNIST-rot-12k compared to ICNN’s 0.98.

This all said, I think the paper is well-written and very clear. The structure is straightforward and the experiments seem repeatable from the descriptions made. The stated aims of the paper are also clear: to learn input transformation invariant CNNs using input-conditioned filters.
Unfortunately a lot of supporting material and prior work has been missed. I list a lot of them here.

Works on input-conditioned filters and invariance. These are the most important

-Dynamic Steerable Frame Networks, Jacobsen et al., 2017
-Dynamic Steerable Blocks in Deep Residual Networks., Jacobsen et al., 2017

Works on input-conditioned filters:

-HyperNetworks, Ha et al., 2016
-Dynamic Filter Networks, de Brabandere et al., 2016

Works on invariance:

-Invariance and neural nets, Barnard and Casasent, 1991
-Group Equivariant Convolutional Networks Cohen and Welling (2015)
-Harmonic Networks: Deep Translation and Rotation Equivariance: Worrall et al. (2017)
-Steerable CNNs, Cohen and Welling (2017)
-Spherical CNNs, Cohen et al. (2018)
-CubeNet: Equivariance to 3D Rotation and Translation, Worrall and Brostow (2018)
-Learning steerable filters for rotation equivariant CNNs, Weiler et al. (2018)
-Gauge Equivariant Convolutional Networks and the Icosahedral CNN, Cohen et al. (2019)

*Questions/notes for the authors*

- Please address the missing references
- Are the input-conditioned filters conditional on position in the activations, or are they shared across all spatial locations of the image? This is not clear from the text.
- The image reconstruction reminds me of Transforming Auto-encoders (Hinton et al., 2016) and Interpretable Transformations with Encoder-Decoder Networks (Worrall et al., 2017). How is your setup different?





**Experience Assessment:**

I have published in this field for several years.

**Review Assessment: Checking Correctness Of Derivations And Theory:**

I carefully checked the derivations and theory.

**Review Assessment: Checking Correctness Of Experiments:**

I carefully checked the experiments.

**Review Assessment: Thoroughness In Paper Reading:**

I read the paper thoroughly.

---

### Official Review · AnonReviewer2 · 2019-10-21
**Official Blind Review #2**

**Rating:** 3

**Review:**

This paper proposed an Input-conditioned Convolutional Neural Network (ICNN) to automatically impose transformation-invariance. The contribution of the manuscript is two-folds
(a) After transforming the input using a pre-determined set of transformations, a set of input-conditioned filter generators are used (and trained) to cater to different input contents.
(b) A decoder is used after the max-pooling layer (of the  Siamese network.) And an L-2 reconstruction loss (with respect to a chosen class representative) is added to the cross-entropy loss for classification.

Overall the paper is well written, and it is fairly easy to read. However, I am not totally convinced that the two contributions of the paper are significant to transformation-invariant representations, and my reasonings are follows

1. Why is a decoder needed in the architecture? If the objective is to achieve transformation-invariance, one can easily compare the L-2 distance between the max-pooled feature maps of a given input to that of the class representative. Why bother using a decoding architecture?
2. Choosing a "class representative" in the CNN seems very restrictive. Why if the underlying task is not image classification? Besides, I am very curious about the experiment on the CIFAR-10 dataset: do the constructed images of all test samples look like the one chosen class representative in the training data? (i.e., compared to figure 3)
3. The input-conditioned filter generation seems a little confusing. Is this what you want to achieve? Say if the pre-determined transformations are rotation (scaling), then the input-conditioned filter should be generated as rotated (scaled) version of the same filters? If so, why not just rotate (rescale) the filters? There are lots of group-equivariant CNNs that have been proposed before for such effect. Besides, I am confused why fractionally-strided convolutions are used for filter generation?

Other comments:
1. The reference for fractionally-strided convolutions should be fixed.
2. Why there is no bias term in convolutional modules (page 4, second paragraph?)
3. What does ICNN short for? The first appearance of the abbreviation in the abstract needs more explanation.

**Experience Assessment:**

I have published in this field for several years.

**Review Assessment: Checking Correctness Of Derivations And Theory:**

I carefully checked the derivations and theory.

**Review Assessment: Checking Correctness Of Experiments:**

I assessed the sensibility of the experiments.

**Review Assessment: Thoroughness In Paper Reading:**

I read the paper at least twice and used my best judgement in assessing the paper.

---

> ### Author Response · Authors · 2019-11-06
> **Justification**
>
> Why decoder is needed?
> -> Dimension of extracted max pool features is different from the representative image.  Therefore, a reconstruction decoder is employed. Another solution would have been to map the representative image to a lower dimension space and use it directly to calculate L-2 distance with max pool features.
> -> Another need for having a decoder is that we can visualize the reconstructed image and check the decoder's validity.
>
> Representative image?
> ->There is no limitation that the representative image should only be a single image. Based on the complexity of the class, multiple representative images can be easily utilized in the network during training time.
> ->The representative image helps the network to extract abstract features from the input image which are necessary to reconstruct a transformation free corresponding image. This idea can be easily extended to text-based models where the representative text can be decided based on the problem statement. (For instance, if we consider the problem of sentiment analysis which takes a text utterance as input and predicts its polarity to be angry, happy, etc., then, in this case, a representative text can be a vector which is close to the input text's corresponding sentiment in the embedding space.)
>
>  If so, why not just rotate (rescale) the filters?
> ->Given an input image, we want our filter generator module to generate the best filters based on the input image rather than explicitly changing the filters or features because during the test time we will not be knowing which transformations should be applied to the extracted features or filters. Therefore, a framework that automatically extracts the best input-conditioned features is needed.
> ->Now the image can have rotation, translation or their combination, it is not feasible to explicitly handle these transformations by employing corresponding transformations in the feature or filter space. Therefore, a framework that automatically extracts the best input-conditioned features is needed.

---

### Official Review · AnonReviewer1 · 2019-11-05
**Official Blind Review #1**

**Rating:** 3

**Review:**

The proposed method in this paper tries to make the CNN robust to the input image transformation by learning to generate convolutional filters.
The proposed architecture has two main parts.
1) Filter Generation:  Given an input image, a set of predefined transformations are applied to the image. After extracting features from the input transformed images with the Siames network, a set of Convolutional filter are estimated. The idea is that these input-dependent filters can compensate all of  transformation in the image.
2) Classification and reconstruction part: The generated convolutional filters are applied to the image and after extracting deeper features a representation vector is computed. This representation vector will be used for classification and reconstruction of the input image to make sure that it has all of the necessary information.

Positive points:
1) The writing is clear.

Negative points:
The proposed method is not novel. The proposed method will be robust to the transformations that are used during training but it cannot generalize to other useen transformations.
2) The experimental results are weak, the authors should compare their method on more difficult datasets like ImageNet dataset.
3) The authors should compare their proposed method with the "Spatial transformer networks, NIPS 2015." in detail.

In conclusion, my recommendation for this paper is "weak reject".


**Experience Assessment:**

I have published in this field for several years.

**Review Assessment: Checking Correctness Of Derivations And Theory:**

I carefully checked the derivations and theory.

**Review Assessment: Checking Correctness Of Experiments:**

I carefully checked the experiments.

**Review Assessment: Thoroughness In Paper Reading:**

I read the paper thoroughly.

---

### Decision · Program_Chairs · 2019-12-19

**Decision:**

Reject

**Comment:**

This paper proposes a CNN that is invariant to input transformation, by making two modifications on top of the TI-pooling architecture: the input-dependent convolutional filters, and a decoder network to ensure fully transformation invariant. Reviewer #1 concerns the limited novelty, unconvincing experimental results. Reviewer #2 praises the paper being well written, but is not convinced by the significance of the contributions. The authors respond to Reviewer #2 but did not change the rating. Reviewer #3 especially concerns that the paper is not well positioned with respect to the related prior work.  Given these concerns and overall negative rating (two weak reject and one reject), the AC recommends reject.